# Fluoride Bio-Sorption Efficiency and Antimicrobial Potency of Macadamia Nut Shells

**DOI:** 10.3390/ma15031065

**Published:** 2022-01-29

**Authors:** Humbelani H. Nekhavhambe, Rabelani Mudzielwana, Mugera W. Gitari, Wasiu B. Ayinde, Oisaemi U. Izevbekhai

**Affiliations:** 1Environmental Remediation and Nano Science Research Group, Department of Geography and Environmental Sciences, Faculty of Science, Engineering and Agriculture, University of Venda, Private Bag X5050, Thohoyandou 0950, South Africa; 11631512@mvula.univen.ac.za (H.H.N.); rabelani.mudzielwana@univen.ac.za (R.M.); twasiu33@gmail.com (W.B.A.); oisaemii@gmail.com (O.U.I.); 2School of Chemistry and Material Science, Technical University of Kenya, Haile Selassie Avenue, P.O. Box 52428, Nairobi 00200, Kenya

**Keywords:** antimicrobial potency, bio-sorption, chemisorption, fluoride, macadamia nut shells

## Abstract

This paper presents the potential application of macadamia nut shells (MNS) in the bio-sorption of fluoride and its antimicrobial potency against common pathogens encountered in surface water resources. The efficiency of MNS in the sorption of fluoride was determined using batch mode experiments, while the antimicrobial potency was investigated using the well disc diffusion assay method. The maximum fluoride sorption capacity of 1.26 mg/g was recorded at an initial fluoride concentration of 5 mg/L, adsorbent dosage of 0.5 g/100 mL, contact time of 120 min and initial pH of 6. The adsorption kinetics data were better described with a pseudo second order model, indicating the dominance of the chemisorption mechanisms. The Langmuir adsorption isotherm model described the isotherm data suggesting a monolayered adsorption. The thermodynamic parameters, ∆*Gº* and ∆*Hº*, confirmed that *F*^−^ sorption by MNS is a spontaneous and endothermic process. The bio-sorbent was regenerated for seven continuous cycles when HCl was used as regenerating solution. The antimicrobial studies revealed that MNS has minimal activity towards *Escherichia coli*, *Staphylococcus aureus* and *Klebsiella pneumoniae*. The MNS showed potentials for application in bio-sorption of fluoride. However, the doping of MNS with metal ions is recommended to enhance its anti-microbial potency.

## 1. Introduction

Rural communities in majority of developing countries rely on groundwater for their domestic uses. However, in many instances, groundwater contains fluoride and microbial contaminants, which poses threat to human health [1] Exposure to fluoride concentration greater than 1.5 mg/L causes dental and skeletal fluorosis, while exposure to pathogens is linked to diarrhoea and cholera amongst other illnesses [2,3]. 

The World Health Organization [4] reported that more than 240 million people worldwide suffer from fluoride related illnesses and about 3.4 million people in developing countries mainly young children die every year from pathogen related diseases. This calls for development of innovative materials that can simultaneously remove fluoride and pathogen from drinking water in line with the sustainable development goals target 6.1, which aims at improving access to clean drinking water by 2030.

A wide range of defluoridation technologies based on the principles of ion-exchange, adsorption, co-precipitation, membrane and electrodialysis techniques have been developed [5] (Garg et al., 2021). However, most of these technologies are more expensive, need trained personnel to operate, generate sludge that needs to be discarded safely and others have shorter life span [6]. 

Adsorption-based techniques are often preferred since they use easily sourced materials, such as clay soils [7], agricultural wastes [8], diatomaceous earth [9], red mud [10], activated carbon [11] and fly ash [12]. Agricultural wastes, such as macadamia nut shells (MNS), consist mainly of cross-linked polymeric chains, such as lignin, cellulose and hemicellulose, with chemically active functional groups that enables them to adsorb various pollutants from water via ion exchange and surface complexation mechanisms [13,14]. 

In addition, they are known to have high surface area and a well-developed microporous structure, which accounts for their higher sorption capacity [15]. The Macadamias South Africa (SAMAC) group indicated that, as of 2019, South Africa was producing about 59,050 tons of macadamia nuts in shell and estimated a further increase in the number of nuts in shell produced to 721,000 tons by 2027 [16].

Pakade et al. [17] evaluated the performance of macadamia nut shells in the bio-sorption of Cr(VI) from aqueous solution and reported a maximum sorption capacity of 45.23 mg/g. They also reported that the material was regenerated for two successive regeneration-reuse cycles. To date, the application of macadamia nut shells towards fluoride removal and their antimicrobial properties towards common pathogens encountered frequently in surface water resources has never been evaluated. This study was therefore developed to evaluate the fluoride removal efficiency as well as the potential application of macadamia nut shells in pathogen removal from water. 

To achieve this target, the following specific objectives were set: (i) to characterize the structural and elemental properties of macadamia nut shells, (ii) to determine the fluoride removal efficiency using batch experiments, (iii) to model the adsorption data using kinetics and isotherm models, (iv) to evaluate the reusability of macadamia nut shells in fluoride removal, and lastly, (v) to determine the anti-microbial potency of the of macadamia nut shells against selected bacteria strains commonly encountered in surface water resources.

## 2. Materials and Methods

### 2.1. Materials

The macadamia nut shells were obtained from Levubu Royal Macadamia farm in Makhado Municipality, Limpopo Province, South Africa. All analytical grade chemicals and reagents were purchased from Rochelle chemicals, Johannesburg, South Africa. This includes sodium hydroxide (NaOH), hydrochloric acid (HCl), potassium chloride (KCl), sodium fluoride (NaF) and Total Ionic Strength Adjustment Buffer (TISAB III).

### 2.2. Preparation and Characterization of Macadamia Nutshell Powder

Macadamia nut shells (MNS) were washed thoroughly to remove dust using Milli-Q water (18.2 MΩ/cm) and then oven dried at 80 °C overnight. Thereafter, dried MNS were pulverized using Retsch planetary ball milling machine PM 400 to pass through <250 µm sieve and stored in a zip locked plastic bag. The elemental composition of the powdered MNS was analysed using Thermo Scientific ^TM^ Flash 2000 Series Organic Elemental Analyser (Thermo Scientific, Cambridge, UK). The functional groups were determined using Bruker Alpha Platinum-ATR Fourier transformation infrared spectroscopy (FTIR). The crystallinity was determined using D8 advanced X-ray diffractometer (XRD) (Bruker) with Cu-Kα Radiation as the source. The surface morphology was determined using scanning electron microscope (SEM) (Leo1450 SEM) (Bruker, Germany). Barrett Joyner Halenda (BJH) and BET isotherms were used to determine the pore characteristics and specific surface area. The pH_pzc_ was determined using a titration-based method [18].

### 2.3. Batch Fluoride Removal Experiments

Batch experiments were carried out to evaluate the performance of MNS in fluoride removal. The effect of contact time and kinetics were determined at initial fluoride concentration of 5 mg/L and adsorbent dosage of 0.5 g/100 mL while varying the agitation time from 5 to 240 min. After each experiment, the samples were filtered, and the residual fluoride concentration was measured using an ion selective electrode calibrated with four standard solutions (0.1, 1, 10 and 100 mg/L) each containing 1 mL of TISAB III solution per 10 mL of standard solution. 

The electrode was attached to the Thermo Scientific Orion Star A211 Benchtop pH/ISE Meter. Same meter was also used for pH measurements. The effect of pH in fluoride removal was determined by varying the initial pH of the solution from 2 to 12 using 0.1 M NaOH and 0.1 M HCl to adjust the pH. Adsorbent mass of 0.5 g was then added to make up dosage of 0.5 g/100 mL, and then the mixtures were agitated for 120 min. The effect of adsorbent dosage was evaluated by varying the adsorbent mass from 0.1 to 0.6 g. The solution containing initial fluoride concentration of 5 mg/L with pH of 6 ± 0.5, and agitation times of 120 min were used. 

The adsorption isotherms were evaluated by varying initial concentration from 5 to 30 mg/L at temperatures of 298, 308 and 318 K. The adsorbent dosage of 0.5 g/100 mL, contact time of 120 min and initial pH of 6 ± 0.5 were used. The influence of individual co-existing ions in fluoride removal were evaluated by agitating a solution containing 5 mg/L of fluoride and 5 mg/L of each of the following ions; SO_4_^2−^, Cl^−^, NO_3−_, CO_3_^2^, Ca^2+^ and Mg^2+^ separately for 120 min. The adsorbent dosage of 0.5 g/100 mL and initial pH of 6 ± 0.5 were used. For accuracy and quality assurance purposes, experiments were carried out in triplicate. Equations (1) and (2) were used to calculate percentage fluoride removal and the adsorption capacity, respectively.
(1)$$\%\;Fluoride\;removal=\left(\frac{C_{0}-C_{e}}{C_{0}}\right)\times 100$$
(2)$$Q=\left(\frac{C_{0}-C_{e}}{m}\right)\times v$$
where *C_o_* and *C_e_* (mg/L) represent the initial and final fluoride ion concentration; *v* (L) is the volume of the solution and *m* (g) is the adsorbent mass.

### 2.4. Regeneration and Reuse Potential of the Adsorbent

The reusability of the adsorbent was evaluated as follows: 0.5 g of MNS loaded with fluoride was reacted with 0.1 M KCl and 0.1 M HCl solutions to desorb the loaded fluoride by shaking for 120 min using reciprocating table shaker. Thereafter, mixtures were filtered, and the residues were collected and rinsed to a neutral pH using Milli-Q water and then oven dried at 110 °C for 8 h. The dried regenerated adsorbent was then reused for fluoride removal. The procedure was repeated for up to six cycles using KCl as regenerant and seven cycles using HCl as regenerant.

### 2.5. Antimicrobial Potency

Antimicrobial activity of the MNS was investigated using the Agar-Well disc diffusion assay method (Kirby–Bauer method). *E. coli*, *S. aureus* and *K. pneumoniae* were used as indicator strains. A solution of 100 mL Mueller–Hinton broth agar was prepared by dissolving 2.1 g of Mueller–Hinton Agar in Milli-Q water. The mixture was autoclaved at 121 °C for 15 min. 

A volume of 5 mL of the prepared broth was pipetted onto clearly labelled 15 mL tubes. Thereafter, 3–4 colonies of *E. coli*, *S. aureus* and *K. pneumoniae* were inoculated onto the tubes and then incubated at 37 °C for a period of 3 h. Subsequently, the cultured bacterial strains were spread onto separate plates containing solidified fresh agar. We pipetted 100 µL of MNS solution on the centre of the plate and incubated at 37 °C for 48 h. Thereafter, the zone of inhibition was recorded to determine the antimicrobial potency of the MNS.

## 3. Results and Discussion

### 3.1. Physiochemical Characterization

#### 3.1.1. Elemental Composition

The elemental composition of MNS powder is presented in Table 1. We observed that MNS is mainly composed of carbon (49.56%), oxygen (44.08%) and hydrogen (6.19%) with traces of nitrogen (0.2%). A similar observation was reported in other lignocellulosic materials, such as walnut shells [19] and raw macadamia nutshell powder [17].

#### 3.1.2. Functional Groups

Figure 1 shows the FTIR spectra of MNS before and after fluoride bio-sorption. A wide transmittance band observed at wavelength regions 3351 cm^−1^ is linked to vibration and stretching of hydroxyl (OH^−^) groups of the moisture absorbed in the cellulose structure [20]. The vibration band at 2928 cm^−1^ wavelength region is linked to aliphatic group (C–H). The band at 1738 cm^−1^ is linked to a carboxylic group (C=O). The bands at 1454 cm^−1^ is assigned to the stretching and vibration of a C–C bond. 

The band at 1248 cm^−1^ indicating the C–OH together with the band of C–O at 1030 cm^−1^ are linked to the vibration and stretching of the phenols, ketones, ethers and esters in the surface of the adsorbent. After defluoridation, there was a decrease in peak intensity and shift in band to 3361, 1741, 1456, 1268 and 1091 cm^−1^. Yang et al. [21] and Albadarin et al. [22] also reported a change in intensities as well as the shifts of absorption bands at 1454 and 1244 cm^−1^, and this was associated to the oxidation of lignin when contacted by the adsorbate.

#### 3.1.3. X-ray Diffraction Analysis

The X-ray diffraction spectra showing of MNS before and after fluoride removal is depicted in Figure 2. Major diffraction peaks are observed at 2θ degree = 17.31°, 22.17° and 34.35°, and they are ascribed to native crystalline cellulose (C_6_H_12_O_6_) structure. The X-ray diffraction spectra of MNS after fluoride removal showed similar peaks suggesting no change in the crystalline structure of the material after exposure to fluoride ions.

#### 3.1.4. Morphological Analysis

The morphology MNS before and after fluoride removal is depicted in Figure 3a,b. The micrograph shows flaky fold-like structures with some crystals on the surface of MNS. The flaky-like structures are linked to crystalline peaks observed in XRD. After fluoride adsorption, the surface appears smooth with some few clusters of agglomerates on top. The change in structure after fluoride removal could be an indication that the material is more saturated with fluoride ions that may have diffused onto the internal layers of the adsorbent.

#### 3.1.5. Surface Area Analysis

Table 2 presents the summary of BET suface area, pore distribution and pore volume results for MNS. We noted that MNS has a surface of 1.69 m^2^/g with an avarage pore diameter and volume of 15.4 nm and 0.01 cm^3^/g, respectively. The pore distribution curve for MNS in Figure 4 shows that the majority pores of the material ranges from microporous to mesoporous range (1 to 15 nm). However, the avarage pore diameter of 15.4 nm suggests dominance of mesoporous structures. Therefore, we assumed that during fluoride sorption, fluoride ions will be able to diffuse from the mesopores of the biosorbent into the micropores.

### 3.2. Batch Adsorption Experiment

#### 3.2.1. Effect of Contact Time

Figure 5 shows the variation of the *F*^−^ adsorption capacity with contact time. The fluoride adsorption capacity was found to be increasing gradually as the contact time increases. This could be due to availability of active sorption sites on the surface of the adsorbents for fluoride ions in the solution. The steep slope between 5 and 40 min could be ascribed to the fast movement of *F*^−^ from the bulk solution onto the adsorbent, while the plateau between 40 and 120 min could be an indication of diffusion of fluoride ions from the mesopores into the micropores of the adsorbent, which is then followed by adsorption within the pores as the time progresses to 180 min. Therefore, 120 min was chosen to be the optimum contact time for subsequent experiments.

The fluoride adsorption data generated at various contact times was fitted to pseudo-first order (PFO) and the pseudo-second order (PSO) reaction kinetics models together with the intra-particle diffusion model to elucidate the mechanism as well as the rate limiting steps for fluoride sorption by MNS. The pseudo-first order suggests that adsorption occurred via physisorption process. Equation (3) shows the nonlinear equation of the PFO model [23].
(3)$$q_{t}=q_{e}\left(1-e^{-k1t}\right)$$

Pseudo second order, on the other hand, assumes that adsorption occurs via chemisorption and involves ion exchange reactions, and it is expressed by the nonlinear Equation (4) [23,24,25].
(4)$$q_{t}=\frac{q_{e}^{2}k_{2}t}{1+k_{2}q_{e}t}$$
where *q_t_* (mg/g) is adsorption capacity at a given time *t* (min), *q_e_* (mg/g) is the maximum sorption capacity at equilibrium and *k*_1_ (min^−1^) and *k*_2_ (g/mg·min) are rate constant parameters for pseudo first and second order models, respectively. The nonlinear plots for PFO and PSO are included in Figure 5, while the respective constant values are shown in Table 3. Based on the correlation coefficient values (R^2^) in Table 3, the adsorption data for fluoride by MNS is described by pseudo second order suggesting chemisorption and ion exchange mechanisms. 

The reaction rate constant *k*_1_ for PFO was found to be 0.35 min^−1^ and, for PSO (*k*_2_), was 1.26 min^−1^ (Table 3) suggesting that chemisorption was much faster and dominant as compared to physisorption. This could have been influenced by the readily available OH^−^ on the surface of the MNS for the exchange with fluoride ions as well as the chemically active functional group, which easily form chemical bonding with fluoride. The fitting of the data into pseudo second order was further confirmed by lower chi-square (X^2^) value obtained (Table 3).

Adsorption is normally a complex process involving movement of adsorbate molecules from the solution into the exterior of the adsorbent particles and further diffuse onto the interior of the adsorbent [26,27]. To further confirm the particle diffusion and strengthen the understanding the rate limiting steps during fluoride adsorption by MNS, intra-particle diffusion model of Weber–Morris, which is depicted by Equation (5), was applied [28].
(5)$$q_{t}=k_{i}t^{0.5}+C_{i}$$
where *q_t_* (mg/g) shows the amount of fluoride ions adsorbed at a given time, *t* (min); *K_i_* (mg g^−1^ min^−1^) is the intra-particle diffusion rate constant. The value of *K_i_* was determined from the slope and intercept of *t^0.5^* vs. *q_t_*. Figure 6 shows the intra-particle diffusion plot for fluoride adsorption by MNS. 

We observed that the plot yielded three clear phases confirming three different process that takes place during adsorption. The first phase is attributed to the adsorption in the external layer of the adsorbent, while the second and third phases are attributed to the diffusion of ions into the interior of the adsorbent and adsorption within the pores of the material, respectively [23,29]. 

During phase 1 fluoride ions are attracted via electrostatic forces to the surface of the adsorbent, which is then followed by the diffusion of adsorbate ions into the mesoporous structures of the MNS (phase 2) where it starts interacting with the atoms of active functional groups within the pores of the adsorbent particles resulting in chemisorption (phase 3). The rate of constant values obtained at different phases are shown in Table 4. We observed that *K_1_* is higher than *K_2_* and *K_3_* indicating that boundary layer adsorption occurred much faster than the subsequent intra-particle diffusion and the adsorption at equilibrium.

#### 3.2.2. Effect of pH

The pH of the solution influences the charges on the surface of the adsorbent and consequently affect the behaviour of fluoride adsorption. Figure 7a show the variations of percentage fluoride removal various initial solution pH levels. The results showed that the percentage fluoride removal decreased as the solution became more acidic and as it became more alkaline. The optimum fluoride removal of 48% was observed at pH 6. To understand the material’s surface charges at various pH levels, the pH point of zero charge (pHpzc) of the material was studied, and the results are presented in Figure 7b. 

From the results, we found that the pHpzc of MNS is ≈7.2 ± 0.5. This implies that, at 7.2 ± 0.5, the material has net charges of zero. Below 7.2 ± 0.5 the surface of the material is positively charged, while above 7.2 ± 0.5, the surface is negatively charged. Therefore, lower percentage fluoride removal at acidic where the surface is positively charged and H^+^ ions dominate the surface could be due to the formation of weak HF acid in the solution leading to lower fluoride removal (Equation (6)). 

The fluoride adsorption at pH below pHpzc negative fluoride ions may be attracted to the positive surface of the adsorbent via electrostatic attraction forces (Equation (7)). At near neutral pH where the optimum uptake occurred, fluoride adsorption could be due to ion exchange and because of weak van der Waal forces (Equation (8)). Conversely, at pH above pHpzc where the surface is negatively charged and OH^−^ dominates the surface, there could be electrostatic repulsion forces formed between negatively charged surface and negative fluoride ion leading to lower fluoride adsorption. However, fluoride could also be adsorbed as a results of ion exchange between the OH^−^ and *F^−^* (Equations (9) and (10)).
(6)$$H^{+}+F^{-}\to HF$$
(7)$$MOH+F^{-}+H_{3}O^{+}\to MOH_{2}^{+}-F^{-}+H_{2}O$$
(8)$$MOH+F^{-}+H_{3}O^{+}\to M^{+}-F^{-}+2H_{2}O$$
(9)$$MOH+F^{-}\to MF+OH^{-}$$
(10)$$M\left(OH\right)2^{-}+2F^{-}\to MF^{-}+2OH^{-}$$
where *M* represent the elements in the adsorbent (MNS) surface.

#### 3.2.3. The Effect of Adsorbent Dosage

Figure 8 shows the changes of percentage fluoride removal and adsorption capacity as a function of adsorbent dosage. The results show that the percentage of fluoride removal increase with adsorbent dosage increases from 0.1 g to 0.5 g/100 mL. Conversely, the adsorption capacity decreases as the adsorbent dosage increases. These trends could be due to increasing the adsorption sites for limited fluoride ions as the dosage increases. An adsorbent dosage of 0.5 g/100 mL was therefore selected as the optimum for subsequent experiments.

#### 3.2.4. Effect of Initial Concentration

The variation of adsorption capacity with equilibrium fluoride concentration at different temperature is depicted in Figure 9. We observed that, at both temperatures, the fluoride adsorption capacity increased with the increasing equilibrium concentration. On contrary, the increase in temperature lead to a decreasing adsorption capacity (Figure 9). This suggests that fluoride adsorption by MNS is more feasible at room temperature. Similar results were observed by Tran et al. [30] who suggested that this tendency could be due to decreased number of active sites responsible for adsorption when the temperature increases.

The commonly used nonlinear equations of Langmuir and Freundlich adsorption isotherms models were applied to explain the relationship between the adsorbate molecule ions and adsorbent’s surface. The Langmuir adsorption isotherm model is expressed by Equation (11). The model assumes monolayer adsorption wherein once the adsorbate molecule is adsorbed on a surface, no other ion may be adsorbed at that adsorption site [31]. The Freundlich adsorption isotherm model, which is expressed by Equation (12), is based on assumption that adsorption occurs on a heterogeneous surface and considers multilayer adsorption [31].
(11)$$q_{e}=\frac{q_{max}K_{L}c_{e}}{1+K_{L}c_{e}}$$
(12)$$q_{e}=K_{f}c_{e}^{1/n}$$
where *C_e_* is the equilibrium concentration (mg/L); *q_e_* is the amount of fluoride ion adsorbed (mg/g); *q_max_* (mg/g) is the maximum saturated monolayer adsorption capacity of the adsorbent; and K_L_ is constant related to the affinity between the adsorbent and the adsorbate (L/mg). *K_f_* (mg/g/(mg/L)*^n^*) is the Freundlich constant related to adsorption capacity and 1/*n* is the dimensionless parameter related to the adsorption intensity and the magnitude of the adsorption driving force [32]. 

Adsorption is favourable when 0 < 1/*n* < 1, irreversible when 1/*n* = 1 and unfavourable when 1/*n* > 1. The nonlinear plots of Langmuir and Freundlich adsorption isotherms models are included in Figure 9, while their constant values of the models are shown in Table 5. The adsorption data at both temperatures yielded higher correlation co-efficient values when fitted to Langmuir model than Freundlich model suggesting monolayered adsorption in which there is no interaction between the adsorbed molecule, and once an adsorbate molecule occupies the site, no further adsorption can occur on that site. This could be an indication that the surface of MNS has fixed number of available sorption sites and that all these sites have equal energy.

To further establish the favourability of the adsorption process, the equilibrium dimensionless parameter, R_L_, which is depicted by Equation (13) was calculated based on Langmuir’s constant related to adsorption affinity (*K_L_*) and the initial concentration (*C_i_*).
(13)$$R_{L}=\frac{1}{1+k_{L}C_{i}}$$

Adsorption is favourable when *R_L_* is <1, linear adsorption if *R_L_* > 1 and irreversible when *R_L_* = 0. All calculated value of *R*_L_ lies between 0 and 1 favourable adsorption at both initial concentrations. To further confirm the fitness of the isotherm model, the chi-square (*X*^2^) value for both models were determined [32]. Based on the chi-square value (Table 5), the smallest values were observed in the Langmuir isotherm, meaning that the fluoride adsorption for both temperatures better fit to Langmuir isotherm. Table 6 compares the maximum adsorption capacity of the evaluated macadamia nut shells with other previously reported bio-sorbents. We observed that the MNS evaluated in this present study has comparable sorption capacity with the some of the previously reported bio-sorbents towards fluoride removal.

#### 3.2.5. Adsorption Thermodynamics

Thermodynamics parameters, such as the Gibbs energy change (∆*G*º), enthalpy change (∆*H*º) and entropy change (∆*S*º), were calculated per Equations (14) and (15) to further establish the adsorption mechanisms for fluoride onto MNS [30].
(14)$$∆G^{ᴼ}=-RT\ln K_{c}$$
(15)$$\ln K_{c}=-\frac{∆H^{0}}{RT}+\frac{∆S^{0}}{R}$$
where ∆*G**º* is the Gibbs free energy change (KJ/mol), *R* is the gas constant with the value of 8.314 J mol^−1^ K^−1^ and T is absolute temperature in Kelvin. *Kc* is the dimensionless equilibrium constant determined from the Langmuir constant (*K_L_*). The parameters of ∆*H**º* (KJ/mol) and ∆*Sº* (KJ/mol) represent the standard enthalpy change and entropy change, respectively, which are determined from slope and intercept of ln*K_c_* against 1/*T*. 

Figure 10 ln*K_c_* against 1/*T* shows the plot, while the constant parameters are presented in Table 7. The results showed negative value of ∆*G**º*, which suggests that fluoride sorption by MNS is feasible and occurs spontaneously at a given temperature. Moreover, the value of ∆*Gº* was observed to be decreasing with increasing temperature suggesting that the degree of spontaneity increases with increases in temperature. 

Similar results were reported by Oyekanmi et al. [37] during the adsorption of congo red by treated durian peels. The ∆*H**º* value was found to be positive, which indicates that the process for sorption of fluoride by MNS is endothermic in nature. An endothermic process suggests chemisorption reactions [30]. This is in line with the findings of adsorption kinetics study, which showed better fittings to pseudo second order model of reaction kinetics. The positive value ∆*S**º* suggesting a dissociative adsorption mechanism as well as the random distribution of fluoride ions on the surface of MNS [30].

#### 3.2.6. The Effect of Co-Existing Ions

Figure 11 depicts the effect of co-existing ions in percentage fluoride removal. From the figure, the presence of magnesium (Mg^2+^), calcium (Ca^2+^), sulphate (SO_4_^2−^), nitrates (NO_3−_), chlorine (Cl^−^) and carbonate (CO_3_^2−^) enhances the percentage fluoride by MNS. Similar effects of co-existing ions were observed by Zhou et al. [37]. This could be an indication that the co-existing ions are creating more active charges on the surface of the adsorbent leading to enhanced percentage fluoride removal. These results indicate that MNS has a better advantage for application in groundwater defluoridation since its effectiveness is not limited by other co-existing ions.

#### 3.2.7. Adsorbent Reusability

Figure 12 shows the change in percentage fluoride removal with continuous regeneration-reuse cycles. The percentage of fluoride removal decreases with continuous reuse of the adsorbent when KCl was used as the regenerant. Quantitatively, the percentage fluoride removal decreased from 42.4% to 15.2%. The decrease could either be due to the gradual dissolution of MNS during treatment of used adsorbent by KCl or due to inadequate reactivation of the adsorbent’s active sites. 

The trend for percentage fluoride removal when HCl was used as regenerant showed an increase in fluoride removal from 46% to 62% after the first cycle of treatment. The percentage fluoride removal remains almost constant until the fourth regeneration-reuse cycle and then started to decrease up to the seventh regeneration-reuse cycle. The increase in percentage fluoride removal by be due to protonation of carbonyl groups following the reaction with HCl, which creates more positive charges on the surface of the MNS and enhances the removal of fluoride via electrostatic attraction (Illustrated in Figure 13). Based on these results, HCl is a better regenerant for MNS.

### 3.3. Antibacterial Activity of MNS

Figure 14 depict the antimicrobial activity of MNS against *Escherichia coli* (a), *Staphylococcus aureus* (b) and *Klebsiella pneumoniae* (c). The results indicate that MNS had limited antibacterial properties against both evaluated antimicrobial strains with an ≈1 mm inhibition zone observed. Due to limited antimicrobial activity, further modification of the MNS by doping with metal species is suggested to further enhance the antimicrobial activity.

## 4. Conclusions

This study evaluated the physicochemical characteristics of MNS and its application towards the removal of fluoride and pathogens from water. From the results, MNS was observed to have potential for fluoride bio-sorption. It was found to be more effective at an initial pH of 6, which is suitable for drinking water. The performance of MNS was enhanced by the presence of co-existing ions, which could be an advantage when treating groundwater containing other co-exiting ions. 

The results showed that MNS can be regenerated for up to seven cycles using 0.01 M HCl. The sorption of fluoride by MNS occurred via chemisorption, and this was a monolayered adsorption as confirmed by the better fitting of the data to the pseudo second order and Langmuir adsorption isotherm model. Thermodynamics studies revealed that fluoride sorption by MNS is a spontaneous process and that fluoride ions are distributed randomly on the surface. 

Antimicrobial studies revealed that MNS has limited potency towards *Escherichia coli*, *Staphylococcus aureus* and *Klebsiella pneumoniae*. Based on these findings, we conclude that MNS can be used for the bio-sorption of fluoride. However, further modifications re suggested to enhance the fluoride sorption capacity and antimicrobial activity towards pathogens commonly found in surface water resources.

## Figures and Tables

**Figure 1 materials-15-01065-f001:**
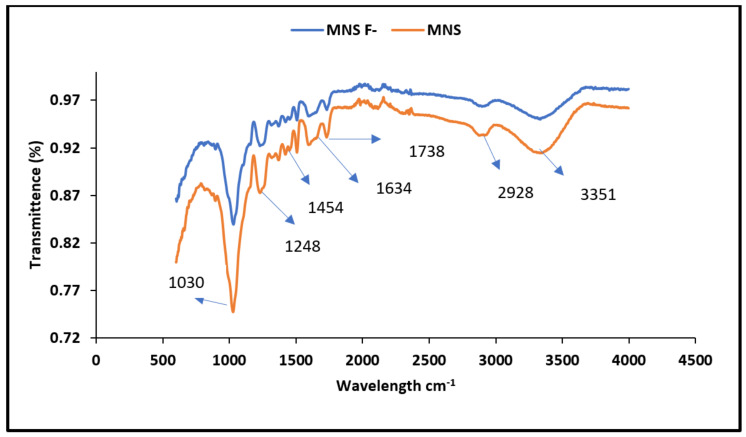
FTIR spectra of macadamia nutshell powder before and after fluoride removal.

**Figure 2 materials-15-01065-f002:**
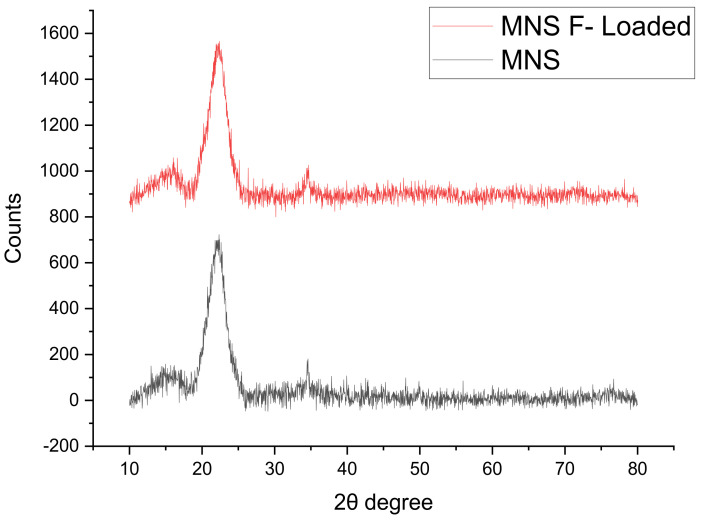
XRD spectra for macadamia nutshell power before and after fluoride removal.

**Figure 3 materials-15-01065-f003:**
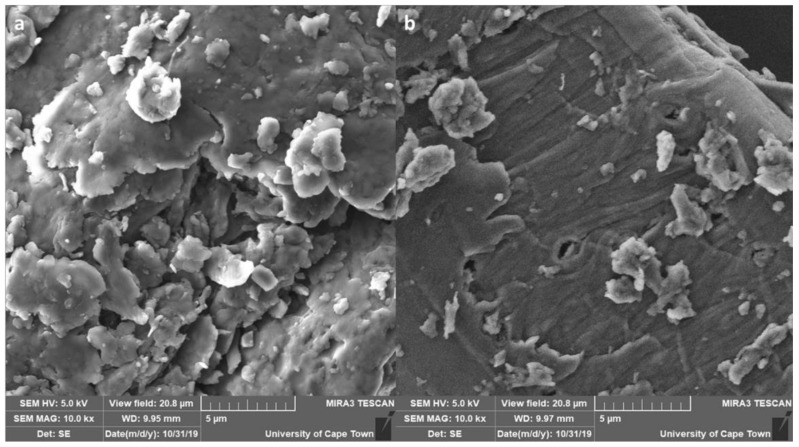
Micrographs of macadamia nut shells before (**a**) and after (**b**) fluoride removal.

**Figure 4 materials-15-01065-f004:**
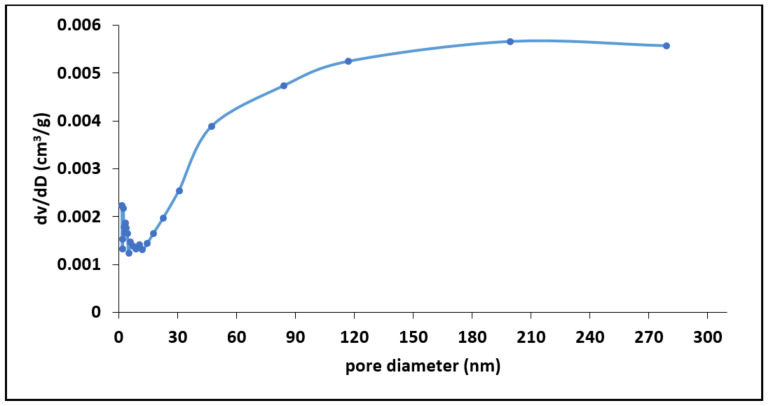
Pore distribution curve for macadamia nut shells.

**Figure 5 materials-15-01065-f005:**
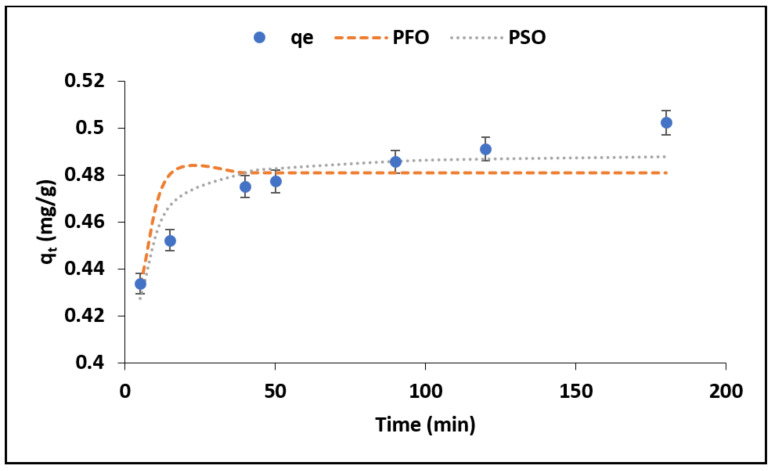
Adsorption capacity and adsorption kinetics for fluoride removal by raw macadamia nutshell powder.

**Figure 6 materials-15-01065-f006:**
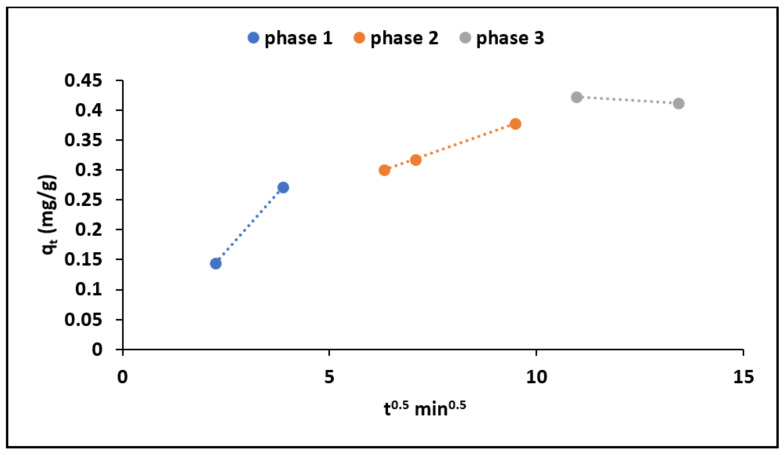
Intra-particle diffusion plot for fluoride adsorption onto MNS.

**Figure 7 materials-15-01065-f007:**
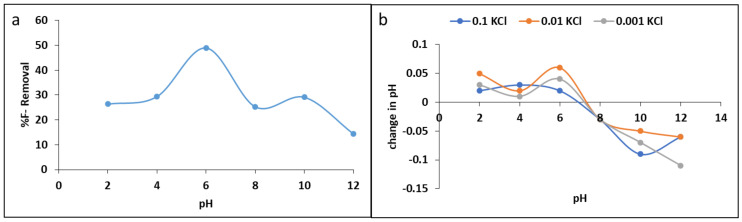
(**a**) Effect of pH on fluoride removal and (**b**) point of zero charge.

**Figure 8 materials-15-01065-f008:**
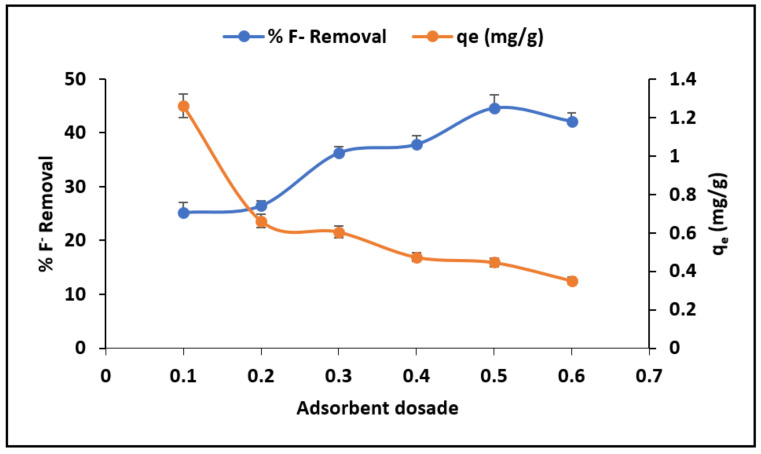
Effect of adsorbent dosage in %Fluoride removal and adsorption capacity.

**Figure 9 materials-15-01065-f009:**
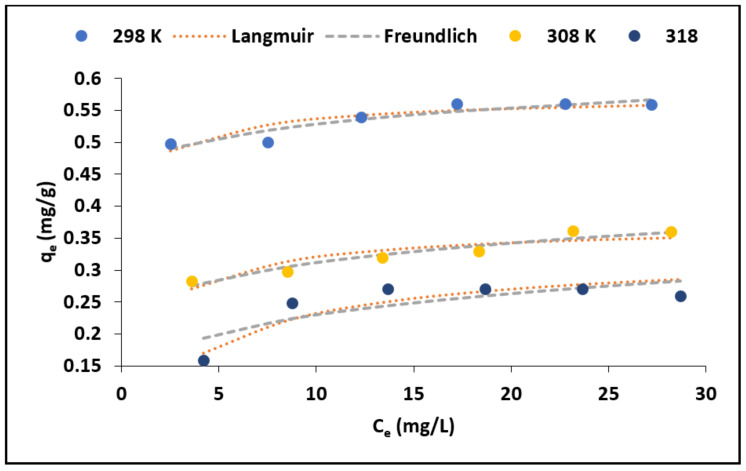
Variation of adsorption capacity and equilibrium fluoride concentration and nonlinear adsorption isotherm plots.

**Figure 10 materials-15-01065-f010:**
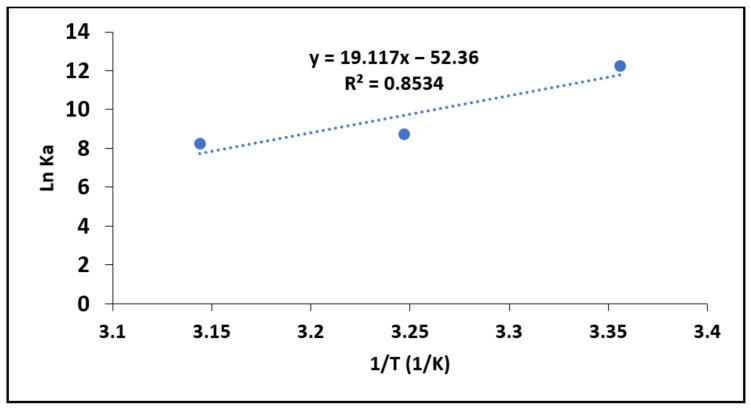
lnKc as a function of reciprocal of adsorption temperatures.

**Figure 11 materials-15-01065-f011:**
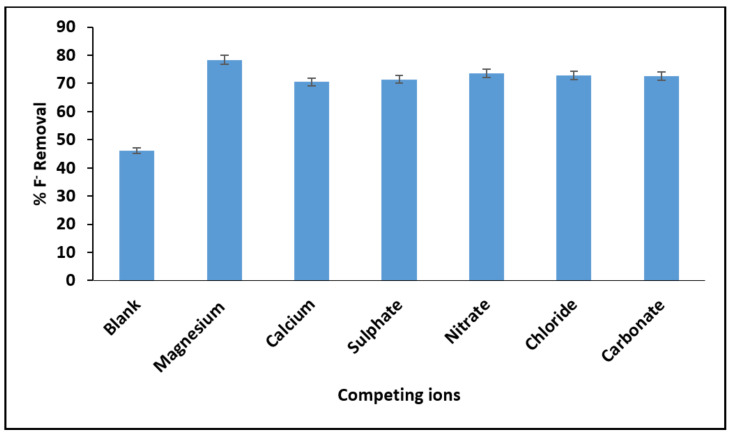
Variation of percentage fluoride removal in the presence of co-existing ions.

**Figure 12 materials-15-01065-f012:**
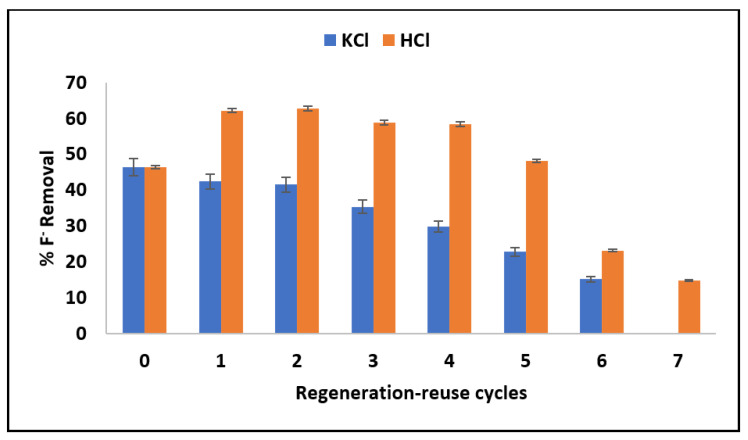
Variation of percentage fluoride removal with a continuous regeneration-reuse cycle.

**Figure 13 materials-15-01065-f013:**
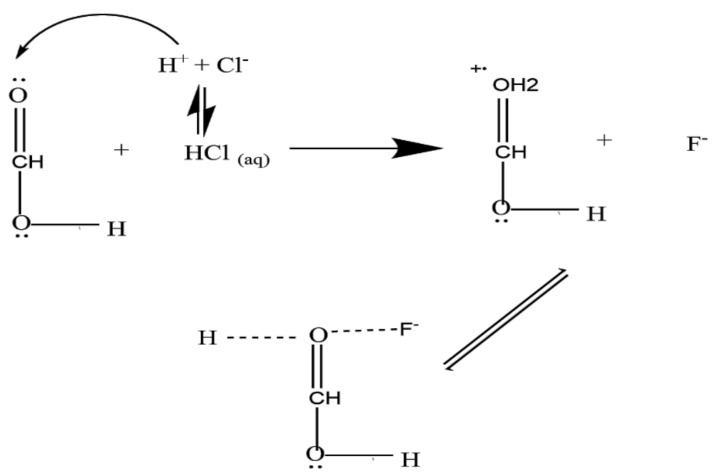
Protonation of the functional groups during regeneration by HCl.

**Figure 14 materials-15-01065-f014:**
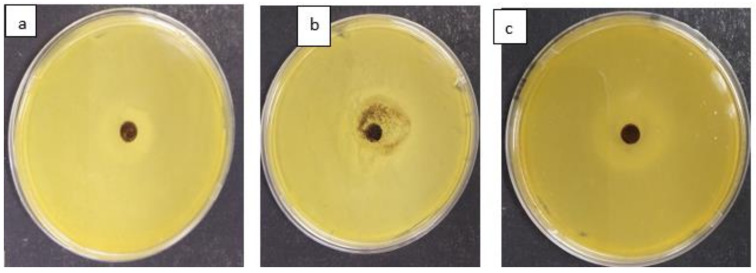
Antimicrobial activity of MNS against (**a**) *Escherichia coli*, (**b**) *Staphylococcus aureus* and (**c**) *Klebsiella pneumoniae*.

**Table 1 materials-15-01065-t001:** Elemental composition of macadamia nutshell powder.

Element/Oxide	Composition (%)
C	49.56
O	44.08
H	6.20
N	0.20

**Table 2 materials-15-01065-t002:** Surface area, pore area and volume of the raw MNS.

Surface Area (m^2^/g)	Pore Diameter (nm)	Pore Volume (cm^3^/g)
1.69	15.4	0.01

**Table 3 materials-15-01065-t003:** The calculated parameters for the pseudo first order and pseudo second order reaction kinetics of raw MNS.

		PFO			PSO		
*K_1_* (min^−1^)	*q_e_* (mg/g)	R^2^	*X* ^2^	*K^2^* (g/mg·min)	*q_e_* (mg/g)	R^2^	*X* ^2^
0.35	0.51	0.89	0.001412	1.26	0.53	0.95	0.000569

**Table 4 materials-15-01065-t004:** Constant values of intra particle diffusion.

Model		MNS
Intra particle diffusion	*K_1_* (mg/g min^−1^)	0.08
	*K_2_* (mg/g min^−1^)	0.02
	*K_3_* (mg/g min^−1^)	0.005
	R^2^ (phase 1)	1
	R^2^ (phase 2)	0.99
	R^2^ (phase 3)	1

**Table 5 materials-15-01065-t005:** Calculated Langmuir and Freundlich isotherm parameters.

	Langmuir Isotherm				Freundlich Isotherm			
Temperature (K)	*q_m_* (mg/g)	*K_L_* (L/mg)	*X^2^*	R^2^	1/n	*K_f_* (mg/g/(mg/L)*^n^*)	R^2^	*X* ^2^
298	0.57	1.10	0.00188	0.99	0.08	0.43	0.86	0.01156
308	0.37	0.33	0.00294	0.99	0.15	0.22	0.99	0.00107
318	0.05	0.20	0.00269	0.98	0.21	0.14	0.96	0.01160

**Table 6 materials-15-01065-t006:** Comparison of the MNS bio-sorption capacity with other previously reported fluoride bio-sorbents.

Adsorbent	Adsorption Capacity (mg/g)	pH	Ref
Sugarcane bagasse	1.2	5.4	[33]
Coconut root	1.75	7	[34]
Sawdust raw	1.73	6	[35]
Wheat straw raw	1.93	6	[35]
Activated bagasse carbon	1.15	6	[35]
*Senna auriculata* L. flower petal biomas	1.29	6–7	[36]
Macadamia nut shells	1.26	6	This study

**Table 7 materials-15-01065-t007:** The adsorption thermodynamic parameters.

∆*G^0^* (kj/mol)	∆*H^0^* (kj/mol)	∆*Sᴼ* (kj/mol)
298 K = −30,393.6 308 K = −22,388 318 K = −21,791	158.93	435.32

## Data Availability

Data available on request.

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
