# Peer review of "Fluoride Bio-Sorption Efficiency and Antimicrobial Potency of Macadamia Nut Shells"

_materials, 2022, doi:10.3390/ma15031065_

Round 1

Reviewer 1 Report

Comment:

The paper discussed about the potential use of macadamia nutshells (MNS) based biosorbents for fluoride removal. The authors carried out kinetics and Isotherms studies. The paper shows an interesting work and it will help other researchers who are working in this area. However, authors should attend to following queries and revisions so that the quality of the manuscript can be strengthened.

  1. Instruments details are missing like pH meter, fluoride concentration measurement etc. please discuss in materials and methods part.
  2. Maximum fluoride sorption capacity is very less? Can you provide a comparative table of other biosorbents applied for fluoride removal?
  3. “Effect of pH” section should discuss before contact time. pH play a crucial role in adsorption study so author should optimize pH first.
  4. Error bar should be included in figures. For example figure 8, one plot has error bars but second plot missing
  5. From the regeneration graph, up to 4th cycle adsorbent materials did not lose capacity but in 5th 6th and 7th it shows substantial decrease in adsorbent capacity then how authors reported that up to 7th cycle can use fluoride removal with continuous regeneration-reuse cycle.
  6. The latest research articles related to this work should be cited and compare the results.

Utilization of Durio Zibethinus peels for Congo red removal from aqueous solution: Statistical optimization and Mechanism Studies, Sustainability 13 (2021) 13264.

Author Response

Instruments details are missing like pH meter, fluoride concentration measurement etc. please discuss in materials and methods part.

Response: The pH and fluoride were measured using Thermo Scientific Orion Star A211 Benchtop pH/ISE Meter.

Maximum fluoride sorption capacity is very less? Can you provide a comparative table of other biosorbents applied for fluoride removal?

Response: Table 6 comparing the bio-sorption capacity of MNS with the capacities of previously reported biosorbents was included in the manuscript.

“Effect of pH” section should discuss before contact time. pH play a crucial role in adsorption study so author should optimize pH first.

Response: We acknowledge the fact raised by the reviewer that pH plays crucial role in the bio-sorption of fluoride. Owing to the fact that the optimum pH was found to be at pH ≈6 to near neutral and that the effect of contact time and all other experiments were done at the same pH level, there is no need to change the order of experiments. Further, the most important information presented from the effect of pH was to bring up the adsorption mechanisms and explaining behaviour of fluoride ions at various pH levels. This information was clearly presented in the study.

Error bar should be included in figures. For example figure 8, one plot has error bars but second plot missing.

Response: We acknowledge the fact raised by the reviewer and we have therefore included the error bars in figures.

From the regeneration graph, up to 4th cycle adsorbent materials did not lose capacity but in 5th 6th and 7th it shows substantial decrease in adsorbent capacity then how authors reported that up to 7th cycle can use fluoride removal with continuous regeneration-reuse cycle.

Response: We appreciate the observation from the reviewer. We reported the that the material can be regenerated for up to 7th cycle because although the percentage of removal dropped following regeneration, the material removed substantial percentage of fluoride ions from the solution.

The latest research articles related to this work should be cited and compare the results.

Utilization of Durio Zibethinus peels for Congo red removal from aqueous solution: Statistical optimization and Mechanism Studies, Sustainability 13 (2021) 13264.

Response: We acknowledge the reference suggested by the reviewer and we have added it to justify the thermodynamic studies.

Oyekanmi, A.A., Ahmad, A., Mohd Setapar, S.H., Alshammari, M.B., Jawaid, M., Hanafiah, M.M., Abdul Khalil, H.P.S. and Vaseashta, A., 2021. Sustainable Durio zibethinus-Derived Biosorbents for Congo Red Removal from Aqueous Solution: Statistical Optimization, Isotherms and Mechanism Studies. Sustainability13(23), p.13264.

Reviewer 2 Report

Please use the MDPI template - I think the formatting is not correct the way it is presented at the moment.
Please provide emails for all co-authors, MDPI will ask you for that prior to publication, so you can save some time here.

I would change the title to "...macadamia nutshell sawdust" - if I understand correctly you using ground nutshells and not the whole shells, so this should be changed since it is very different
I would do the same thing in the whole manuscript - you can just call it macadamia nutshell sawdust (MNS) and still use your abbreviation

So you used fluoride desolved in high purity water in your experiments? or did you also used actual wastewater samples?

I think it is UN SDG 6 - there is no 6.1 as far as I know

line 41: "need"

For your literature review/introduction there are three studies I would like you to read and consider.
I think this would be helpful.

Two reviews on bio-sorption with agricultural biomass:
https://doi.org/10.1016/j.molliq.2019.111684
https://doi.org/10.1016/j.jhazmat.2008.06.042

and a recent publication on using other nutshells for biosorption:
https://doi.org/10.1016/j.mineng.2021.107085

The last study is particularly interesting since they propose using the nutshells after sorption for heat generation.
I think the study is very similar to what you did here (targeting a different pollutant). So you could provide an outlook how the material could also subsequently be used for heat generation.
___

How many tons of nutshells does the farm produce per year? this could be relevant if you really want to upscale this
How did you pulverize the nutshells, please elaborate

Once you introduced an abbreviation such as MNS you can and should only use the abbreviation...
I don't really see a significant difference in the XRD spectra - consider removing the figure or explain it better - in India we don't show them usually

Figure 3 - this is not sharp - see if you can get a sharp picture here and make sure the scale can be read

Fig. 4 - use smaller points so that the area from 0-30 can be understood and read clearly

The analysis is sound  -this is how we usually conduct these experiments as well

Fig. 11 - this is really interesting - usually additional ions reduce the sorption efficiency - can you find studies that have a similar outcome to your work?

Chapter 3.3 is interesting but I feel a bit short given the prominence it has in the abstract. Maybe you can explain the importance of this a bit better and provide a little bit of an economic outlook here, or how this could really be used in rural commmunities in RSA.

Author Response

Please use the MDPI template - I think the formatting is not correct the way it is presented at the moment.
Please provide emails for all co-authors, MDPI will ask you for that prior to publication, so you can save some time here.

Response: These details about other authors were captured in the online form during submission.

I would change the title to "...macadamia nutshell sawdust" - if I understand correctly you using ground nutshells and not the whole shells, so this should be changed since it is very different
I would do the same thing in the whole manuscript - you can just call it macadamia nutshell sawdust (MNS) and still use your abbreviation

So you used fluoride dissolved in high purity water in your experiments? or did you also used actual wastewater samples?

Response: Synthetic water was used for most of the experiments. However, knowing that groundwater contains other co-existing ions, we also evaluated the effectiveness of the material in the presence of common co-existing ions.

I think it is UN SDG 6 - there is no 6.1 as far as I know.

Response: The UN SDG 6 is broken down into several sections with 6.1 being the one aiming at provision of clean water. In this context, we referred to target 6.1.

line 41: "need"

Response: change was effected

For your literature review/introduction there are three studies I would like you to read and consider.
I think this would be helpful.

Two reviews on bio-sorption with agricultural biomass:
https://doi.org/10.1016/j.molliq.2019.111684
https://doi.org/10.1016/j.jhazmat.2008.06.042

and a recent publication on using other nutshells for biosorption:
https://doi.org/10.1016/j.mineng.2021.107085

The last study is particularly interesting since they propose using the nutshells after sorption for heat generation.
I think the study is very similar to what you did here (targeting a different pollutant). So you could provide an outlook how the material could also subsequently be used for heat generation.

Response: We thank the reviewer for suggesting these significant references and some of them were are now included in the paper for improvement. The main aim of this study was to evaluate the applicability of material in fluoride removal. Hence we feel looking at how material could be used in heat generation could be deviating from the scope of the current paper. However, this suggestion will be very useful in future studies looking and post adsorption treatment of the material.

Febrianto, J., Kosasih, A.N., Sunarso, J., Ju, Y.H., Indraswati, N. and Ismadji, S., 2009. Equilibrium and kinetic studies in adsorption of heavy metals using biosorbent: a summary of recent studies. Journal of hazardous materials162(2-3), pp.616-645.

Anastopoulos, I., Pashalidis, I., Hosseini-Bandegharaei, A., Giannakoudakis, D.A., Robalds, A., Usman, M., Escudero, L.B., Zhou, Y., Colmenares, J.C., Núñez-Delgado, A. and Lima, É.C., 2019. Agricultural biomass/waste as adsorbents for toxic metal decontamination of aqueous solutions. Journal of Molecular Liquids295, p.111684.

How many tons of nutshells does the farm produce per year? this could be relevant if you really want to upscale this.

Response: According to Macadamias South Africa group (SAMAC), as of 2019 South Africa was producing a total of 59 050 tons of macadamia nuts in shell.  

How did you pulverize the nutshells, please elaborate.

Response: Retsch planetary ball milling machine PM 400 was used to pulverize the nutshells.

Once you introduced an abbreviation such as MNS you can and should only use the abbreviation...
I don't really see a significant difference in the XRD spectra - consider removing the figure or explain it better - in India we don't show them usually.

Response: We thank the reviewer to pointing out this. However, we felt that it is essential to also analyse the crystalline nature of the material since this is known to influence the surface area and pore structure which in turn affect he adsorption properties. Hence this was included in the draft manuscript.

Figure 3 - this is not sharp - see if you can get a sharp picture here and make sure the scale can be read.

Response: We acknowledge the point raised by the reviewer, however, the images we presented are of the best quality we received from the analysist. We hope this could be acceptable.

Fig. 4 - use smaller points so that the area from 0-30 can be understood and read clearly

Response: We have tried to reduce the size of the points to three to make sure points are visible enough.

The analysis is sound  -this is how we usually conduct these experiments as well.

Response: Thank you for pointing out this and acknowledging the analysis.

Fig. 11 - this is really interesting - usually additional ions reduce the sorption efficiency - can you find studies that have a similar outcome to your work?

Response: Zhou et al. (2021) also reported similar effect of co-existing ions.

Chapter 3.3 is interesting but I feel a bit short given the prominence it has in the abstract. Maybe you can explain the importance of this a bit better and provide a little bit of an economic outlook here, or how this could really be used in rural commmunities in RSA.

Response: This section was carried out to evaluate the potency of the material towards removal of pathogens from groundwater. This was in line with the goal of developing a multifunctional adsorbent that can be used for both fluoride and pathogen simultaneously. This at the end will reduce the cost of treating water since one may have to apply one material to treat water unlike having to apply different materials in a case where both pollutants are available in water.

Round 2

Reviewer 2 Report

This is fine with me now. The authors incorporated most changes I requested and addressed the other issues I raised appropriately.

This manuscript is a resubmission of an earlier submission. The following is a list of the peer review reports and author responses from that submission.